# GeromiRs Are Downregulated in the Tumor Microenvironment during Colon Cancer Colonization of the Liver in a Murine Metastasis Model

**DOI:** 10.3390/ijms22094819

**Published:** 2021-05-01

**Authors:** Daniela Gerovska, Patricia Garcia-Gallastegi, Olatz Crende, Joana Márquez, Gorka Larrinaga, Maite Unzurrunzaga, Marcos J. Araúzo-Bravo, Iker Badiola

**Affiliations:** 1Computational Biology and Systems Biomedicine Group, Biodonostia Health Research Institute, C/Doctor Beguiristain s/n, 20014 San Sebastián, Spain; daniela.gerovska@biodonostia.org; 2Computational Biomedicine Data Analysis Platform, Biodonostia Health Research Institute, C/Doctor Beguiristain s/n, 20014 San Sebastián, Spain; 3Department of Cell Biology and Histology, Faculty of Medicine and Nursing, University of Basque Country (UPV/EHU), 48940 Leioa, Spain; patricia.garcia@ehu.eus (P.G.-G.); olatz.crende@ehu.eus (O.C.); joana.marquez@ehu.eus (J.M.); 4Department of Nursing I, Faculty of Medicine and Nursing, University of the Basque Country (UPV/EHU), 48940 Leioa, Spain; gorka.larrinaga@ehu.eus; 5Department of Physiology, Faculty of Medicine and Nursing, University of the Basque Country (UPV/EHU), 48940 Leioa, Spain; 6BioCruces Health Research Institute, 48903 Barakaldo, Spain; 7Centro Salud Legazpi OSI Goierri-Urola Garaia-Osakidetza, 20230 Legazpia, Spain; maite.unzurrunzagaaltube@osakidetza.eus; 8IKERBASQUE, Basque Foundation for Science, C/María Díaz Harokoa 3, 48013 Bilbao, Spain; 9CIBER of Frailty and Healthy Aging (CIBERfes), 28029 Madrid, Spain; 10TransBioNet Thematic Network of Excellence for Transitional Bioinformatics, Barcelona Supercomputing Center, 08034 Barcelona, Spain; 11Computational Biology and Bioinformatics Group, Max Planck Institute for Molecular Biomedicine, Röntgenstr. 20, 48149 Münster, Germany; 12Nanokide Therapeutics SL, Zitek Ed, Rectorado Bajo, Bº Sarriena sn, 48940 Leioa, Spain

**Keywords:** colorectal cancer, liver metastasis, miRNA, tumor microenvironment, geromiRs, histone modifications

## Abstract

Cancer is a phenomenon broadly related to ageing in various ways such as cell cycle deregulation, metabolic defects or telomerases dysfunction as principal processes. Although the tumor cell is the main actor in cancer progression, it is not the only element of the disease. Cells and the matrix surrounding the tumor, called the tumor microenvironment (TME), play key roles in cancer progression. Phenotypic changes of the TME are indispensable for disease progression and a few of these transformations are produced by epigenetic changes including miRNA dysregulation. In this study, we found that a specific group of miRNAs in the liver TME produced by colon cancer called geromiRs, which are miRNAs related to the ageing process, are significantly downregulated. The three principal cell types involved in the liver TME, namely, liver sinusoidal endothelial cells, hepatic stellate (Ito) cells and Kupffer cells, were isolated from a murine hepatic metastasis model, and the miRNA and gene expression profiles were studied. From the 115 geromiRs and their associated hallmarks of aging, which we compiled from the literature, 75 were represented in the used microarrays, 26 out of them were downregulated in the TME cells during colon cancer colonization of the liver, and none of them were upregulated. The histone modification hallmark of the downregulated geromiRs is significantly enriched with the geromiRs *miR-15a*, *miR-16*, *miR-26a*, *miR-29a*, *miR-29b* and *miR-29c*. We built a network of all of the geromiRs downregulated in the TME cells and their gene targets from the MirTarBase database, and we analyzed the expression of these geromiR gene targets in the TME. We found that *Cercam* and *Spsb4*, identified as prognostic markers in a few cancer types, are associated with downregulated geromiRs and are upregulated in the TME cells.

## 1. Introduction

Cancer is a myriad of different diseases acting in a distinct manner dependent on the organ of origin, the cell type from which it was formed and the heterogeneity of each human body [1]. Although two different cancer cells created from the same cell type can develop in a different manner, the most effective way to classify cancer types is to consider their organ of origin. According to such a classification, colorectal cancer (CRC) is one of the most common in both men and women [2]. Usually, CRCs arise from the colon or the rectum mucosal epithelium in the form of a polyp. A few cells that form a part of the mass can evolve into malignant cells and enter the bloodstream. Most of the cells that reach the blood vessels die but a small subpopulation that survives is able to colonize other organs in a process called metastasis. Interestingly, there is an important tropism between the cancer cell type and the metastasized organs [3] according to the so-called “seed and soil” theory of Steven Paget published in 1889 [4]. CRC is not an exception and most of the CRC metastatic cells colonize the liver. The parenchymal cells of the liver, namely, hepatocytes, account for 60–70% of the total liver cells whereas the non-parenchymal cells include cholangiocytes (2–3%), liver sinusoidal endothelial cells (LSECs) (15–20%), Kupffer cells (15%) and hepatic stellate cells (HSCs), also known as Ito cells (5–8%) [5]. In the liver, the metastatic cell interacts with different cell types, changes their phenotype and creates a propitious environment called the tumor microenvironment (TME) [6]. The first interaction of the tumor cell is with the LSECs; i.e., the cells that cover the lumen of the small capillaries called sinusoids. The tumor cell is anchored by the LSECs using different integrins such as VCAM-1 and VLA-4 [7,8]. The tumor then passes through the endothelial barrier and colonizes the parenchyma. During this process, the HSCs, perisinusoidal fat-storing cells and retinoic acid-storing cells located in the perisinusoidal space of the Disse, are activated, transforming into a fibroblastic phenotype [9]. The active fibroblastic HSCs produce a fibrotic extracellular matrix and inflammatory signals promoting tumor development and subsequent activation and recruitment of Kupffer cells [10]. The activation of the TME cells has been broadly studied and many proteins have been described as key factors during the phenotypic transformation as b-FGF, PDGF or TGF-β1 [11]. In addition to the protein signals, epigenetic processes are also involved in the TME transformation as microRNA (miRNA) deregulation [12,13]. MiRNAs are small non-coding RNAs that act as a posttranscriptional gene regulation inhibiting protein expression. MiRNAs have been described as playing a role in many processes such as development [14], cell to cell communication [15] and diseases such as Alzheimer’s [16], cardiovascular dysfunctions [17] and cancer [18]. It is becoming common to classify miRNAs according to their involvement in processes and diseases, e.g., as oncomiRs [19], miRNAs upregulated in cancer cells and inhibiting tumor suppressor genes, metastamiRs, miRNAs that have been shown to initiate invasion and metastasis by targeting multiple proteins [20] or geromiRs [21], a growing list of miRNAs involved in the ageing process. In this work, we identified downregulated geromiRs in LSECs, HSCs and Kupffer cells of the liver TME after colon cancer colonization (Figure 1). For the first time, these geromiRs have been described with their associated aging hallmarks in the context of CRC liver TME.

## 2. Results

### 2.1. MiRNAs in the Liver TME Are Mainly Downregulated

First, we calculated the differentially expressed miRNAs (DEMs) between the TME cells and the control cells. For this analysis, on the one hand, we averaged the expression of the controls of the three studied liver cell types: endothelial (E), Ito (I) and Kupffer (K), and, on the other hand, we averaged the expression of the same three liver cell types in the TME. We observed that the downregulated miRNAs in the TME outnumbered the upregulated miRNAs in the TME (Figure 2, Figure 3 and Figure 4), with the scatterplot in Figure 2A showing the dispersion of the miRNA expression between the control and the TME, and the volcano plot in Figure 2B showing the statistical significance of this dispersion. The DEMs and their expression across the analyzed samples of the control and the TME cells are presented in the heatmap in Figure 3 for downregulated miRNAs, and in Figure 4 for upregulated miRNAs, separately for endothelial, Ito and Kupffer cells. Additionally, for the sake of comparison, we included in the heatmaps samples from CRC primary (TP) and liver metastasis (TM) cells. We ranked the DEMs in order of their statistical significance in Figure 5, where we could see that there was a higher number of downregulated miRNAs (71) compared with the number of upregulated miRNAs (30).

### 2.2. GeromiRs Are Significantly Downregulated in the TME

We observed that the most statistically significant miRNA downregulated in the TME cells, *miR-146a,* was a geromiR (Figure 5A). Liu et al. [22] found that the relative expression level of *miR-146a* in the peripheral blood of colon cancer patients was significantly lower than that of a control group with benign colitis. Simanovich et al. [23] showed that *miR-146a-5p* functions as a control switch between angiogenesis and cell death, its neutralization can manipulate the crosstalk between tumor cells and macrophages, and profoundly change the TME. The second-ranked statistically most significant downregulated miRNA, *miR-7a*/*miR-7*, also a geromiR, inhibits colorectal cancer cell proliferation and induces apoptosis by targeting XRCC2 [24]. The third-ranked most statistically significant miRNA downregulated in the TME cells was yet another geromiR, *miR-21a*, which was in contrast with an observation that the expression level of *miR-21* in lung adenocarcinoma cells, squamous cell carcinoma cells and lung cancer tissues was significantly higher than that of human bronchial epithelial cells and adjacent tissues [25]. Additionally, among the most statistically significant miRNAs downregulated in the TME cells, we found other geromiRs such as *miR-22*, *miR-16*, *miR-15a*, *miR-26b*, *miR-29a* and *miR-20a* among others in decreasing order of statistical significance. Motivated by this discovery, we compiled a comprehensive list of geromiRs and their corresponding functions from several publications [26,27,28] (Table 1). We then searched for which of these geromiRs were represented in our Agilent miRNA microarrays (Table 2). We used this table to analyze the expression of all of the geromiRs in our samples (Figure 6). We then calculated the intersection of these geromiRs with the DEMs found in Section 2.1 (Figure 7A). Interestingly, we found only downregulated geromiRs, and more precisely, 26 downregulated geromiRs with a statistically significant enrichment of *p*-value = 2.746 × 10^−6^, in the TME cells. No geromiRs were upregulated in the TME cells. The expression of the downregulated geromiRs in the TME, together with their functions, are shown in Figure 7B and the distributions of their expression are shown in the violin plots in Figure 7C. The most statistically significant upregulated miRNA in the TME (Figure 5B) was *miR-5132*, which is a tailed miRtron, a class of miRNAs generated through non-canonical pathways; in this case, generated from the end regions of longer introns [29]. The second-ranked statistically significant upregulated miRNA in the TME, *miR*-*1896* (Figure 5B), is known, together with *miR*-*409*-*3p*, to co-operatively participate in IL-17-induced inflammatory cytokine production in astrocytes and the pathogenesis of a multiple sclerosis mouse model via targeting SOCS3/STAT3 signaling [30]. The third-ranked most statistically significant miRNA upregulated in the TME cells (Figure 5B), *miR*-*1306,* is known to target FBXL5 to promote the metastasis of a hepatocellular carcinoma through suppressing the snail degradation [31].

### 2.3. The GeromiRs Downregulated in the TME Are Significantly Involved in Histone Modifications and DNA Methylation

To analyze whether the geromiRs downregulated in TME cells have a potential biological function, we analyzed the statistical enrichment of all downregulated geromiRs across the hallmarks of aging, as shown in Table 2. We found that histone modification and DNA methylation were the top statistically significant functionally enriched categories (Figure 8A). We then checked the gene expression of histone deacetylases (HDACs) and of Sirtuins across all of the analyzed liver samples. With the exception of *Hdac7* and *Sirt3*, we did not find any upregulation of histone modifier-related genes in the TME cells. The expression of the DNA methyltransferases in our samples also did not give conclusive results about dysregulation (Figure 8C).

### 2.4. Cercam and Spsb4 Upregulated in the TME Cells Were Identified as an Element of the Network of Downregulation in the GeromiRs of TME Cells and Their Gene Targets

Identifying miRNA targets is the first and foremost task toward understanding miRNA functions. We employed miRNet 2.0 [32] to find our geromiR gene interaction network (Figure 9A). We then checked the expression of the identified by the network genes, and found two genes upregulated in the TME cells compared with the control, namely, *Cercam* and *Spsb4* (Figure 9B). *CERCAM* (cerebral endothelial cell adhesion molecule) has a low cancer specificity, has been detected in all cancer tissues and is known as an unfavorable prognostic marker in renal cancer, urothelial cancer and ovarian cancer [33]. Very few studies on the function and mechanism of CERCAM in colon and rectal cancers have been published. Based on an analysis of the upregulated genes in the TME that were evaluated through a Kaplan–Meier survival analysis using colorectal cancer data from TCGA, *CERCAM* was found to be associated with a poor survival and epithelial to mesenchymal transition (EMT) [34]. *CERCAM* is an element of the prognostic signature of six differentially expressed and survival-associated genes in all of the three immune subtypes, named the High-Immunity Subtype, Medium-Immunity Subtype and Low-Immunity Subtype, in colon and rectal cancer (*CERCAM*, *CD37*, *CALB2*, *MEOX2*, *RASGRP2* and *PCOLCE2*) identified by a multivariable COX analysis [35]. SPSB4 (a SPRY domain-containing SOCS box protein) is enriched in gliomas and testis cancer tissues and is a favorable prognostic marker in gliomas [33]. The *Cercam*-associated geromiRs found with miRNet 2.0 were *miR-10a*, *miR-26a* and *miR-26b*. *MiR-10a* expression promotes the metastatic behavior of pancreatic tumor cells and its repression is sufficient to inhibit invasion and metastasis formation [36]. The downregulation of *miR-10a* in chronic myeloid leukemia CD34+ cells increases USF2-mediated cell growth [37] and increases the cisplatin resistance of lung adenocarcinoma circulating tumor cells via targeting PIK3CA in the PI3K/Akt pathway [38]. *MiR-26a* can function not only as tumor suppressor but also as an oncogenic miRNA; its function is tumor type-specific and highly dependent on its targets in different cancer cells. Therefore, targeting *miR-26a* in metastatic cancers requires a discreet consideration [39]. Normal human colon cells express low levels of LEF1 and high levels of miR-26b, while human colon cancer cells have a decreased *miR-26b* expression and an increased LEF1 expression. It has been demonstrated that *miR-26b* expression is a potent inhibitor of colon cancer cell proliferation and significantly decreases LEF1 expression [40].

*SPSB4* was identified among five genes, *ADAMTS2*, *HOXA1*, *PCDH10*, *SEMA5A* and *SPSB4,* that were highly/frequently methylated in liquid biopsies of pancreatic cancers although with a relatively low potential compared with the other four markers. In tissue samples, *SPSB4* was identified as a potential DNA methylation marker of pancreatic cancer with the highest AUC (area under the receiver operating characteristic curve) [41]. The *Spsb4*-associated geromiRs found with miRNet 2.0 were *miR-15a*, *miR-16* and *miR-24*. MiRNAs encoded by the *miR-15/16* cluster are known to act as tumor suppressors; the expression of these miRNAs inhibits cell proliferation, promotes the apoptosis of cancer cells and suppresses tumorigenicity both in vitro and in vivo, targeting multiple oncogenes including BCL2, MCL1, CCND1 and WNT3A while the downregulation of these miRNAs has been reported in chronic lymphocytic lymphoma, pituitary adenomas and prostate carcinomas [42]. *MiR-24* was reported to be downregulated in a few types of cancer, indicating its role as a tumor suppressor and reported to be upregulated in a few other types of cancer, even in the same type of cancer, suggesting its role as an oncogene [43].

## 3. Discussion

Since Hayflick and Moorhead [44] formulated the “Hayflick limit” as the finite capacity of human somatic cells to replicate mainly due to the progressive shortening of telomeres during each DNA replication step [45], much research has been focused on cell biology and senescence, and the comparison with the tumor cell creation and perpetuation processes [46,47,48]. In this sense, tumor and senescence have been compared to the two faces of a coin. Both cancer and senescence have similar deregulated processes but while senescence could be considered as a kind of protection mechanism of cells in order to avoid abnormalities, tumor formation could be a deregulated senescence that bypasses a few of the strategies driving the cell to a quasi-quiescent condition. There is evidence of links between the two processes related to biochemical pathways, metabolic processes and epigenetic mechanism, and miRNAs as epigenetic factors are one of the key regulators studied in both processes. MiRNAs, small molecules of RNA that repress protein translation via the RNA messenger block, have been broadly studied as possible disease markers or therapeutic targets. Among the several families of miRNAs, there is one that has been related to ageing and is known as geromiRs [26,27,28]. GeromiRs represent a huge family of miRNAs that participate in different ageing processes such as inflammaging, cell senescence, DNA damage response or mitochondrial dysfunction, all of them key processes in cancer formation. The study of geromiRs and cancer-related processes is not new, anyway, in this study, we focused on new unstudied settings for geromiRs, the tumor microenvironment (TME) cells, i.e., the cells surrounding the tumor and recruited by cancer cells to support their growth and progression. The TME is a crucial component of tumor progression but the phenotypic transformations of the TME cells are less known than those of tumor cells. We compared the miRNAs deregulated in the TME in a murine liver metastasis model with healthy controls and identified considerable geromiR downregulation during the protumoral transformation of liver TME cells. Among the downregulated miRNAs was *miR-22*, whose expression is known to decrease in favorable tumor microenvironments that promote the epithelial to mesenchymal transition and cancer stem-like phenotypes. The downregulation of *miR-22* has been described as being accompanied with the upregulation of many stem cell markers such as *Oct4* and *CD133*, and the downregulation of E-cadherin [49]. E-cadherin is an adhesion molecule downregulated in tumor metastasis [50]. In general, *miR-22* is considered a metabolic silencer and tumor suppressor. *MiR-22* expression levels analyzed in different types of cancer in comparison with normal specimens using information from the TCGA Data Portal were reduced in a hepatocellular carcinoma (HCC), a breast invasive carcinoma and a lung squamous cell carcinoma. The expression level of *miR-22* was inversely associated with the depth of the HCC invasion; furthermore, it was found to be positively correlated with overall survival and disease-free survival [51]. *MiR-29* is another cancer protecting geromiR and its downregulation promotes metastasis and immune microenvironment remodeling in breast cancer via the regulation of LOXL4 [52]. *MiR-15a/16* is an important tumor suppressor gene cluster with a variety of factors that regulate its transcriptional activity and has been shown to play a role in the tumor environment [53]. *MiR-138* was found to be downregulated in human colorectal cancer tissues and cell lines, and the downregulation of *miR-138* was associated with lymph node metastasis and distant metastasis, and was found to always predict a poor prognosis [54]. The *miR-143/-145* cluster is greatly reduced in several cancers, including colon cancer. Both *miR-143* and *miR-145* have been shown to possess anti-tumorigenic activity with an involvement in various cancer-related events such as proliferation, invasion and migration. A statistically significant downregulation (*p*-value < 0.001) of *miR-143* expression levels has been reported in colon tumor samples [55]. The APCMin/+ (multiple intestinal neoplasia) mouse model harbors a germline mutation in the APC tumor suppressor gene and exhibits multiple tumors in the small intestine and colon [56]; the downregulation of *miR-194* promoted tumor formation in APCMin/+ mice [57]. In our study, we found an important set of geromiRs, which were downregulated in the murine model of the CRC TME in the liver. The role of the members of this set of geromiRs in cancer has been confirmed in the literature in tumoral processes related to the inhibition of tumor progression pathways in a few cases or directly as TME cell switching elements in other cases. Furthermore, another important finding of this work is the clustering of downregulated geromiRs in function of their involvement in cellular processes. The downregulated geromiR cluster with largest number of members is related to histone modification and is an example of double epigenetic regulation. It is a cluster of miRNAs acting on epigenetic enzymes. DNA methyltransferase (DNMT) and histone deacetylase (HDAC) inhibition have been reported as anti-angiogenic agents via clusterin, fibrillin 1 and quiescin Q6 modulation [58]. In general, miRNAs and HDACs have a complex relationship that is not yet fully understood but could be critically important. MiRNAs are able to regulate HDACs and influence histone acetylation, while HDACs themselves can regulate miRNA expression. Thus, a careful balance between the two is important to maintain appropriate levels of each in the cell. Interestingly, HDAC inhibitors can alter the expression profiles of miRNAs in cancer and in the brain [59,60,61]. We found that *Hdac7* (ranked second) and *Sirt6* are among the 20 top-ranked differentially expressed genes (DEGs) in Ito cells upregulated in the CRC TME compared with healthy controls [62]. *Hdac7* and *Sirt6* belong to the Class II and Class III families of HDACs. Silencing multiple protein deacetylases is a mechanism by which *miR-22* has an anti-cancer effect. HDAC1 as a *miR-22* target uncovered colon cancer cells [63]. Another important downregulated group of miRNAs are related to Sirtuins. SIRT6 works in multiple molecular pathways related to aging including DNA repair, telomere maintenance, glycolysis and inflammation [64]. SIRT6 is member of the mammalian Sirtuin family of proteins, which are homologous to the yeast Sir2 protein. Sirt6 knock-out mice, in which the gene encoding Sirt6 has been disrupted, exhibit a severe progeria or premature aging syndrome, characterized by spinal curvature, greying of the fur, lymphopenia and low levels of blood glucose [65]. In summary, the liver TME cells show a downregulation of geromiRs that promote cancer progression and the phenotypic transformation of TME cells to cancer-supporting-like cells. Additionally, the most important downregulated geromiRs are clustered in the epigenetic modulation groups such as histone modification, DNA methylation and Sirtuin modulation. MicroRNAs upregulated during the aging process are downregulated in the TME, once again showing the similarities, though in reverse way, between aging and cancer. 

## 4. Materials and Methods

### 4.1. Animals

Balb/c mice (6- to 8-week-old males) were obtained from Charles River Laboratories Spain SA (Barcelona, Spain). Mice were kept in the animal facility of EHU/UPV and had access to standard chow and water ad libitum.

### 4.2. Colorectal Cancer Cells

Murine CRC C26 cells (ATCC, Manassas, VA, USA) syngeneic with Balb/c mice were grown under standard conditions in an RPMI medium (Sigma–Aldrich, St. Louis, MO, USA) supplemented with 10% fetal bovine serum (FBS), penicillin (100 U/mL), streptomycin (100 mg/mL) and amphotericin B (0.25 mg/mL), all purchased from Life Technologies, Carlsbad, CA, USA.

### 4.3. Control and Tumor-Activated Hepatic Cell Isolation and Culture

Control and tumor-activated primary cultures of hepatic cells, LSECs, Ito cells and Kupffer cells were isolated from livers with CRC metastasis or from healthy controls. Balb/c mice were anesthetized with isofluorano and underwent surgical incisions on their left broadside. Mice were inoculated into the spleen at the incision sites using 2 × 106 of C26 colon carcinoma cells. The control mice were inoculated with PBS. Fourteen days later, all mice were sacrificed and the liver cells were removed and purified by differential centrifugation. In brief, the mice were perfused with Clostridium histolyticum collagenase P (Sigma–Aldrich, St. Louis, MO, USA) through the cava vein and the obtained cell suspension was twice centrifuged, resulting in a parenchymal (PC)-enriched pellet and a non-PC-enriched supernatant. The non-PC-enriched supernatant was layered on Percoll gradients (25% on top of 50% *w*/*v*) to obtain LSECs and on Percoll gradients (33% on top of 50% *w*/*v*) to obtain Ito cells. After centrifugation, the interphase between the two density cushions was collected and contained purified non-PC-enriched LSECs in the first gradient and Ito cells in the second one. Both solutions contained Kupffer cells, which were further separated by adherence assays. All cell fractions were washed and cultured with an RPMI 1640 medium (Sigma–Aldrich, St. Louis, MO, USA) supplemented with 10% FBS and used in different experiments a maximum of 24 h after isolation (Life Technologies, Carlsbad, CA, USA).

### 4.4. Omics Analysis

One-hundred ng of total RNA from each fraction was labeled and hybridized onto Agilent mouse RNA and miRNA microarrays, Release 18.0 (Agilent Technologies, Santa Clara, CA, USA) following the standard Agilent protocol with RNA and miRNA complete labelling and hybridization kits including Agilent RNA spike-ins. The results were scanned using an Agilent G2565CA microarray scanner. Scanned TIFF image files were processed using Agilent feature extraction software (v10.7.3.1) to extract the raw data. Briefly, samples were shortly separated in a polyacrylamide gel and proteins we re-reduced with dithiothreitol, alkylated with iodoacetamide and digested with trypsin. After extraction, the peptides were finally resuspended in 0.1% trifluoroacetic acid. Approximately 500 ng of peptides were subsequently separated on an Ultimate 3000 rapid separation liquid chromatography system (Thermo Fisher Scientific, Dreieich, Germany) and analyzed on an Orbitrap Elite (Thermo Fisher Scientific, Dreieich, Germany) hybrid mass spectrometer. All omics data were normalized with the quantile method. To find the statistically significant molecules and their GO enrichment, we used the method described previously [62] with the selection threshold θ_DEG_ = θ_DEM_ = 4 and significance threshold α_DEG_ = α_DEM_ = 0.001. For each mRNA sample, we used four biological replicates; for each miRNA sample we used three biological replicates except two for ECs and KTs.

## Figures and Tables

**Figure 1 ijms-22-04819-f001:**
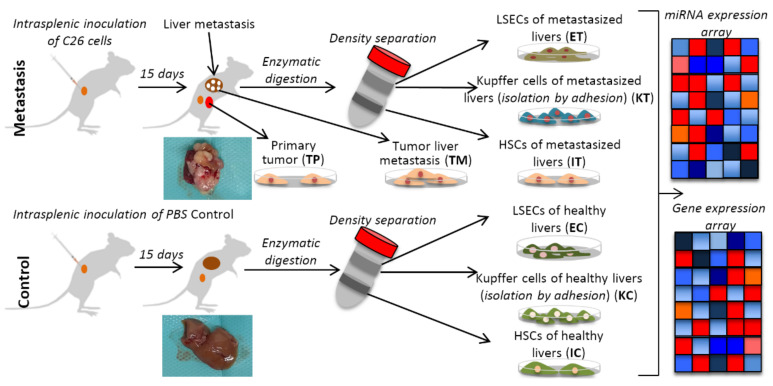
Schematic representation of the experimental design with the murine liver metastasis model. Two groups of mice were inoculated, one with C26 murine colon cancer cells (Tumor) and another with PBS (Control), respectively. After 15 days, the mice were perfused and after Percoll gradient centrifugation, three types of liver cells were collected, namely, liver sinusoidal endothelial cells (E), Ito cells (I) and Kupffer cells (K) from the healthy control (C) and the tumor microenvironment, TME (T), and isolated to perform omics experiments (gene expression and miRNA expression microarrays). TP and TM denote colorectal cancer (CRC) primary and tumor liver metastasis cells, respectively. Insert images of metastasized and healthy livers harvested in our experiments are shown for each condition.

**Figure 2 ijms-22-04819-f002:**
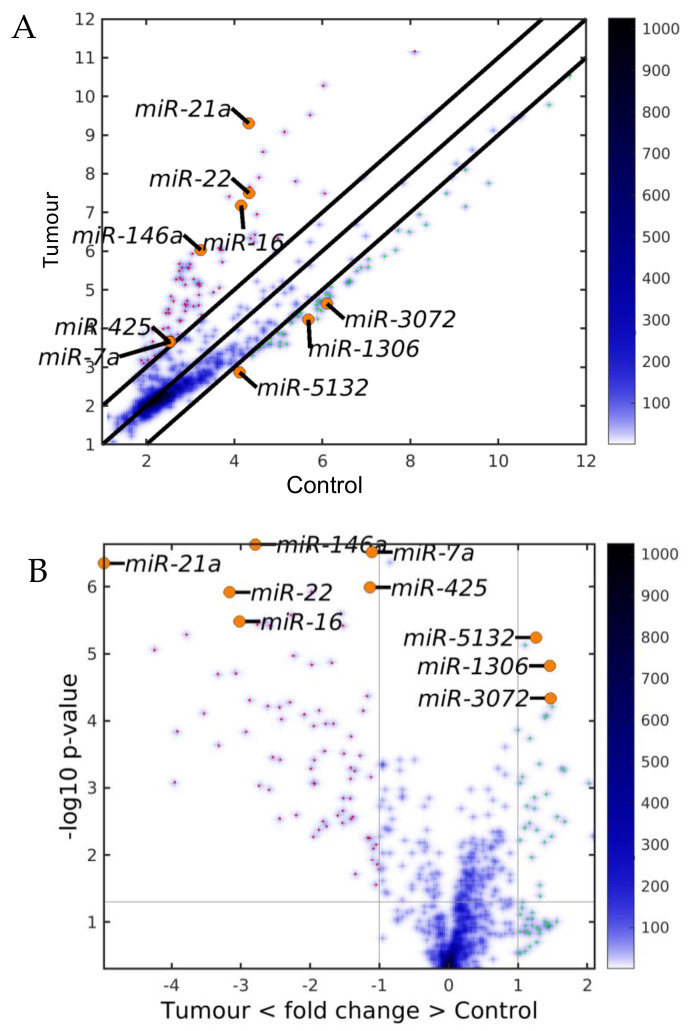
Differentially expressed miRNAs (DEMs) between the control and the tumor microenvironment (TME) cells. (**A**) Scatter plot and (**B**) Volcano plot. The black lines are the boundaries of the two–fold changes in the levels between the mean expression of the three studied liver cell types in the control and the TME samples. The miRNAs upregulated in the TME samples (ordinate) are shown with red dots, and those downregulated, with green. Several geromiR positions are shown as orange circles. The color bar indicates the scattering density. Darker blue color corresponds to higher scattering density. The expression levels are log_2_–scaled.

**Figure 3 ijms-22-04819-f003:**
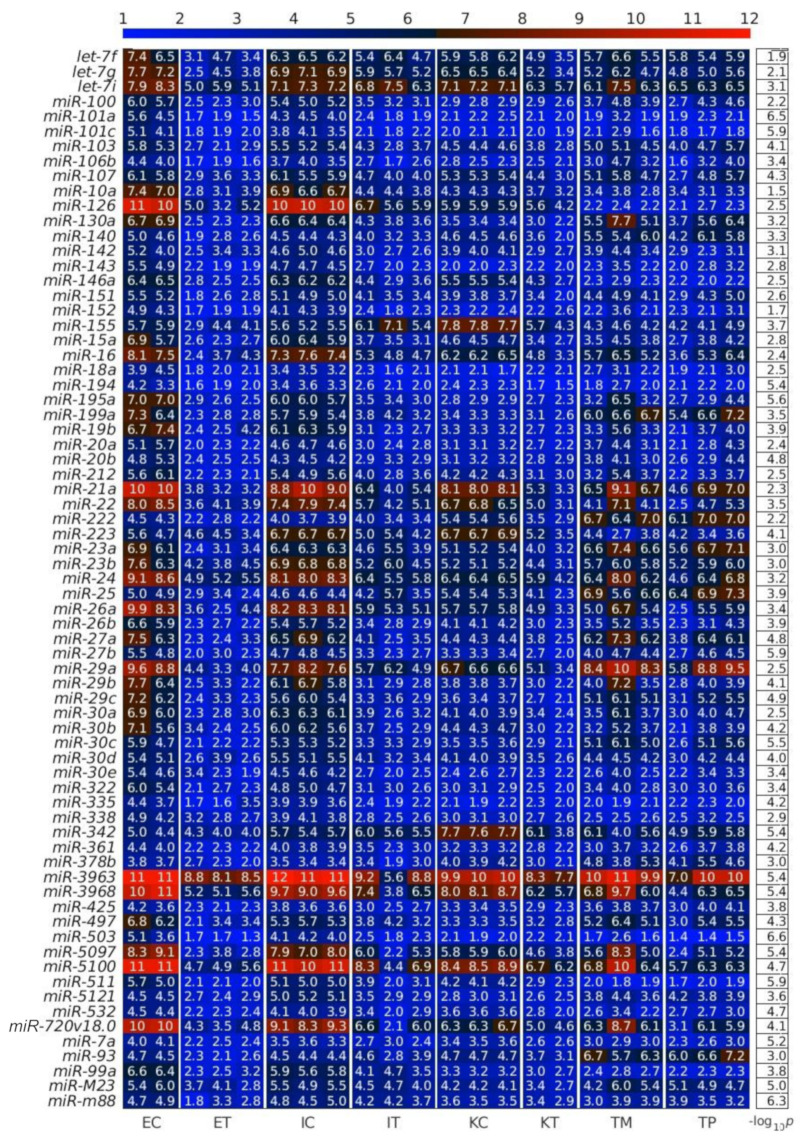
Downregulated miRNAs between the control and the tumor microenvironment (TME) cells. Heatmap of the expression of the differentially expressed miRNAs (DEMs) in lexicographic order of the miRNA names. The color bar codifies the miRNA expression in log_2_ scale. Higher miRNA expression corresponds to redder color. The −log_10_ (*p*-value) of the DEMs are presented in a table to the right of the heatmap. The samples are denoted with E, I and K for endothelial, Ito and Kupffer cell, C and T for the control and the TME cells, and TP and TM for colorectal cancer (CRC) primary and liver metastasis cells, respectively.

**Figure 4 ijms-22-04819-f004:**
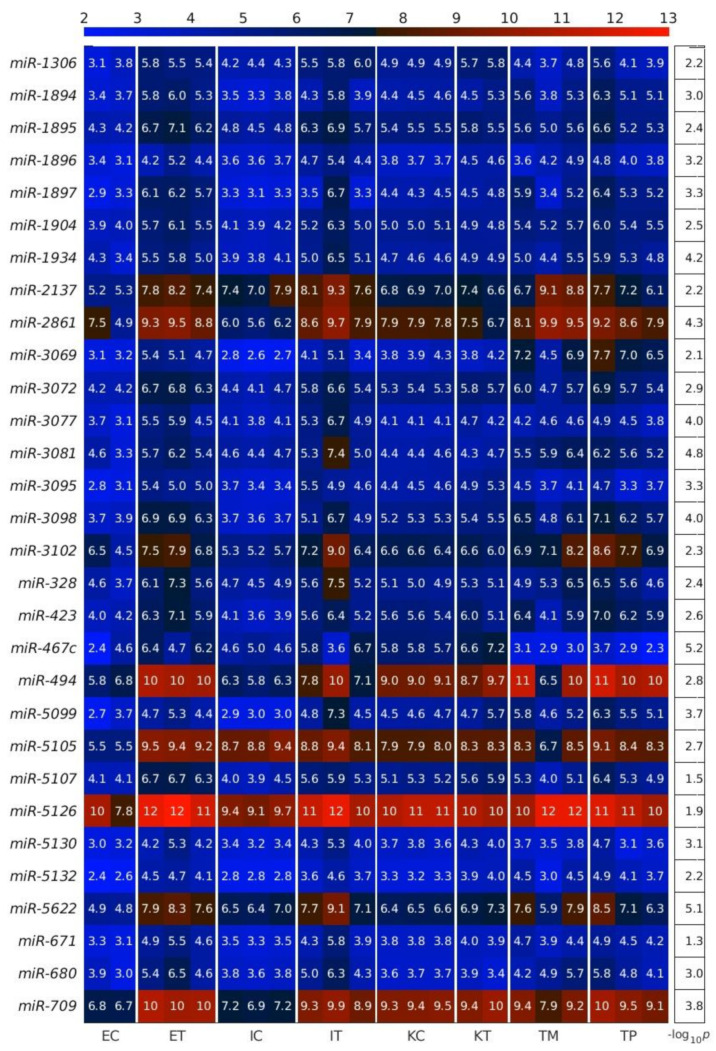
Upregulated miRNAs between the control and the tumor microenvironment (TME) cells. Heatmap of the expression of the differentially expressed miRNAs (DEMs) in lexicographic order of the miRNA names. The color bar codifies the miRNA expression in log_2_ scale. Higher miRNA expression corresponds to redder color. The −log_10_ (*p*-value) of the DEMs are presented in a table to the right of the heatmap. The samples are denoted with E, I and K for endothelial, Ito and Kupffer cells, C and T for the control and the TME cells, and TP and TM for colorectal cancer (CRC) primary and liver metastasis cells, respectively.

**Figure 5 ijms-22-04819-f005:**
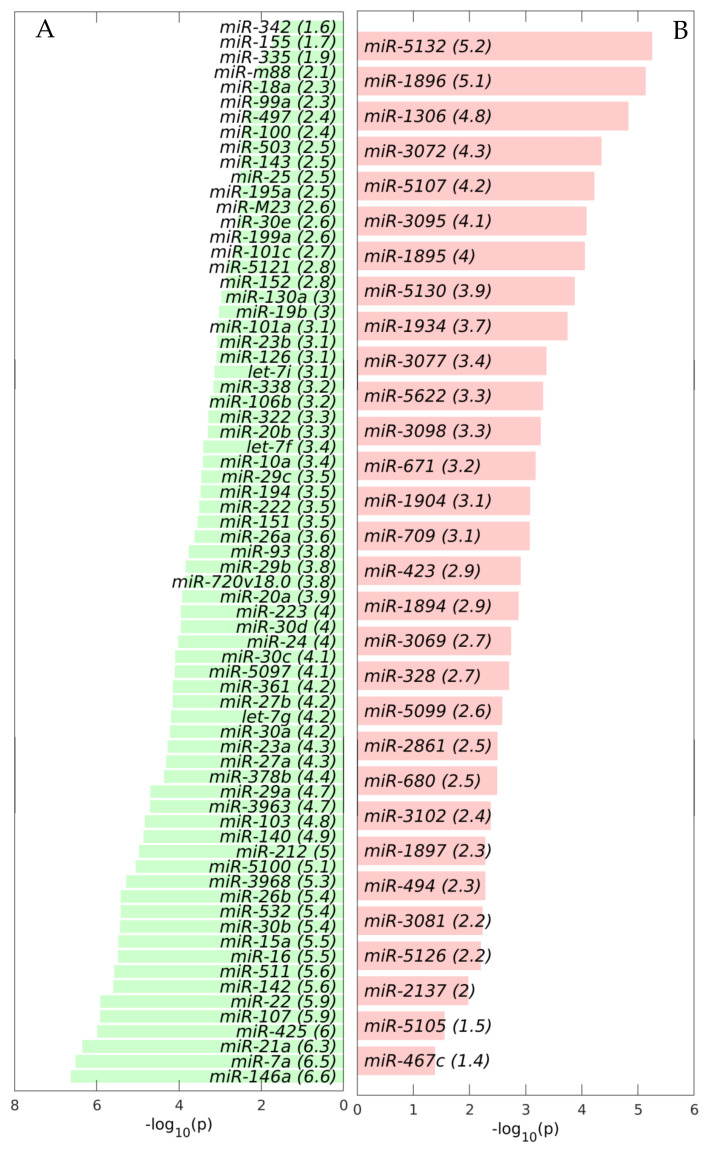
Differentially expressed miRNAs (DEMs) between the control and the tumor microenvironment (TME) cells. Bar plots of the −log_10_ (*p*) of the statistical significance of the DEMs. (**A**) Downregulated in the TME DEMs. (**B**) Upregulated in the TME DEMs.

**Figure 6 ijms-22-04819-f006:**
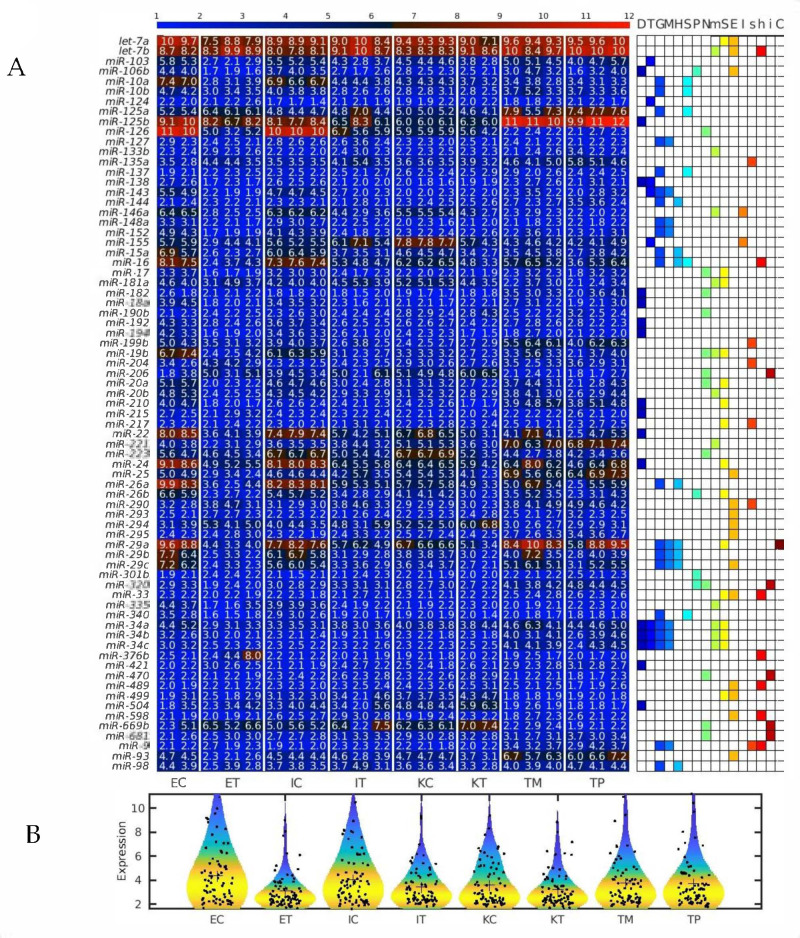
Several microRNAs (miRNAs) downregulated simultaneously in the three types of tumor microenvironment (TME) cells are hallmarks of eukaryotic aging. (**A**) Heatmap of all geromiRs and their annotation with different aging hallmarks. D: altered DNA damage response, T: loss of telomeres, G: changes in gene regulation, M: DNA methylation, H: histone modifications, S: regulation of splicing, P: changes to protein homeostasis, N: altered nutrient sensing, m: mitochondrial dysfunction, S: cellular senescence, E: stem cell exhaustion, I: inflammaging, s: Sirtuins, h: stem cell homeostasis, i: insulin/IGF1, C: altered intercellular communication. The samples are denoted with E, I and K for endothelial, Ito and Kupffer cells, and C and T for the control and the TME cells, respectively. TP and TM denote colorectal cancer (CRC) primary and liver metastasis cells, respectively. (**B**) Violin plots of the expression distribution of the miRNAs associated with the hallmarks of eukaryotic aging. The crosses represent the position of the means, while the black points represent the spread of the expression of the miRNAs used to build the distributions.

**Figure 7 ijms-22-04819-f007:**
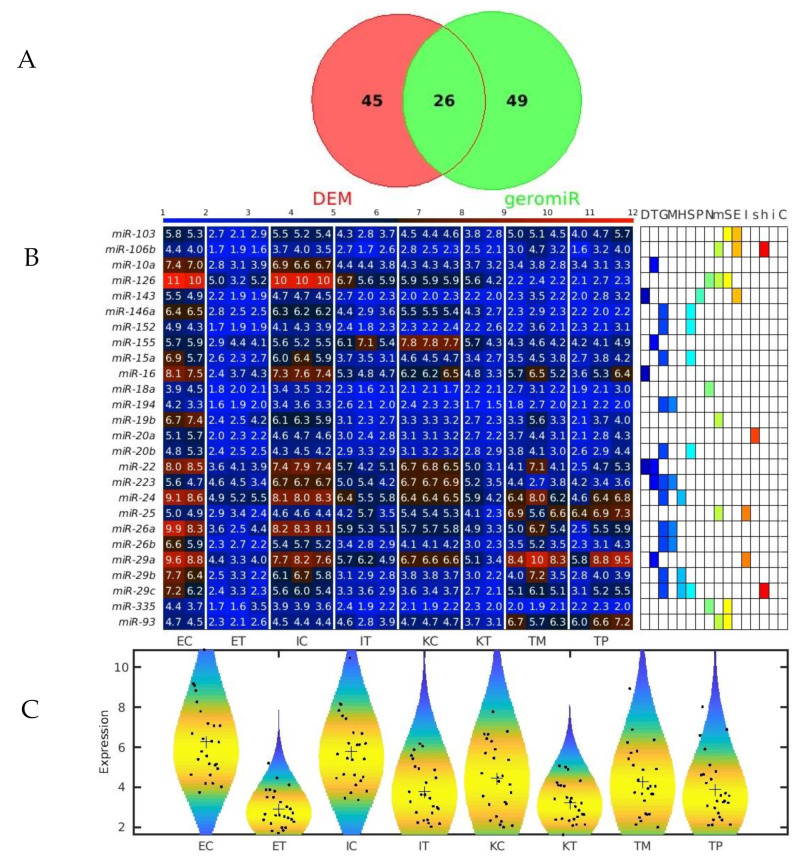
GeromiRs downregulated in the TME. (**A**) Venn diagram of the intersection between the downregulated differentially expressed miRNAs (DEMs) in the colorectal cancer (CRC) liver tumor microenvironment (TME) cells and the geromiRs. (**B**) Heatmap of aging-related miRNAs and their annotation with different aging hallmarks. D: altered DNA damage response, T: loss of telomeres, G: changes in gene regulation, M: DNA methylation, H: histone modifications, S: regulation of splicing, P: changes to protein homeostasis, N: altered nutrient sensing, m: mitochondrial dysfunction, S: cellular senescence, E: stem cell exhaustion, I: inflammaging, s: Sirtuins, h: stem cell homeostasis, i: insulin/IGF1, C: altered intercellular communication. The samples are denoted with E, I and K for endothelial, Ito and Kupffer cells, and C and T for the control and the TME cells, respectively. TP and TM denote the CRC primary and liver metastasis cells, respectively. (**C**) Violin plots of the expression distribution of the miRNAs associated with the hallmarks of eukaryotic aging. The crosses represent the position of the means, while the black points represent the spread of the expression of the miRNAs used to build the distributions.

**Figure 8 ijms-22-04819-f008:**
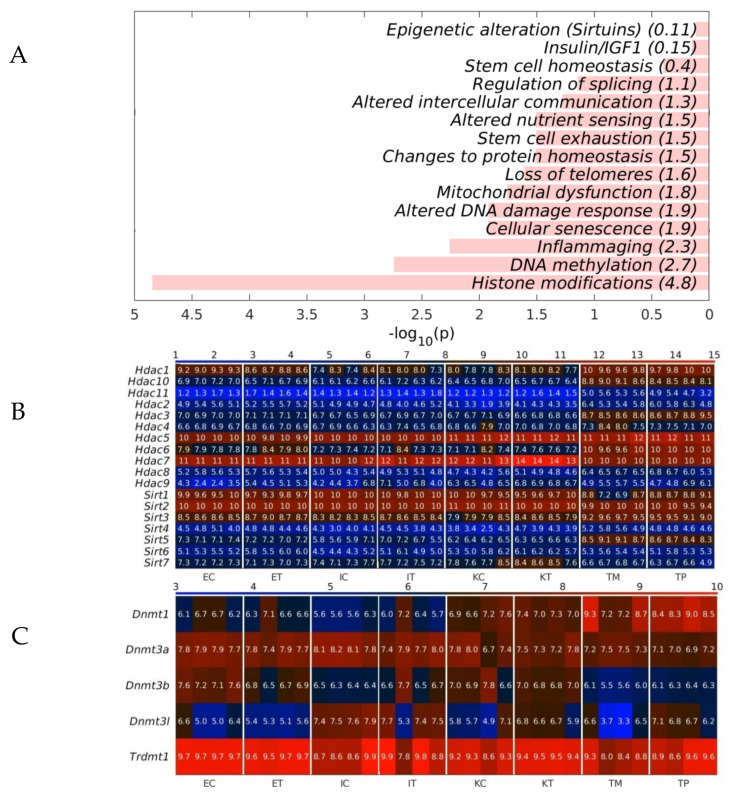
Statistical enrichment of the hallmarks of the geromiRs downregulated in the tumor microenvironment (TME). (**A**) Bar plots of the −log_10_ (*p*) of the statistical enrichment of the hallmarks of the geromiRs downregulated in TME. (**B**) Heatmap of the expression of the histone deacetylases (HDACs) and of the Sirtuins. (**C**) Heatmap of the expression of the DNA methyltransferases. The color bars codify the gene expression in log_2_ scale. Higher gene expression corresponds to redder color. The samples are denoted with E, I and K for endothelial, Ito and Kupffer cells, C and T for the control and the TME cells and TP and TM for the CRC primary and liver metastasis cells, respectively.

**Figure 9 ijms-22-04819-f009:**
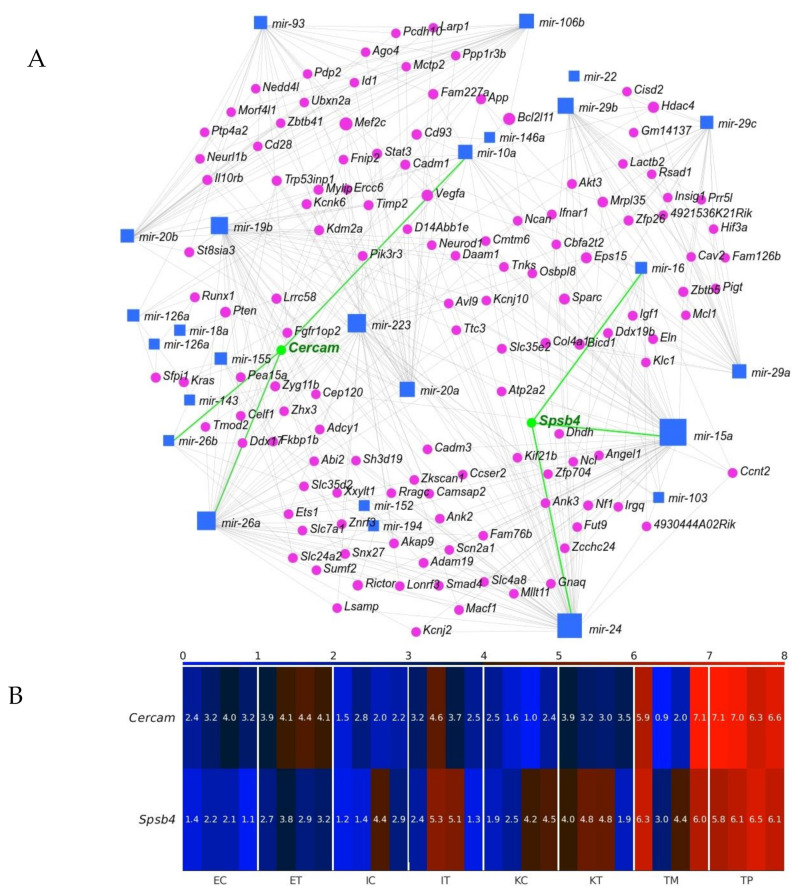
Interaction between geromiRs downregulated in the tumor microenvironment (TME) cells and their gene targets. (**A**) Network of the geromiRs downregulated in the TME cells and their gene targets. The blue squares mark geromiRs and the magenta circles mark gene targets. The gene names in green, *Cercam* and *Spsb4*, mark the gene targets upregulated in the TME cells in relation to the healthy control cells, and the green lines, their connections with their associated geromiRs. (**B**) Heatmap of the expression of the gene targets upregulated in the TME cells. The color bar codifies the miRNA expression in log_2_ scale. Higher miRNA expression corresponds to redder color. The samples are denoted with E, I and K for endothelial, Ito and Kupffer cells, C and T for the control and the TME cells, and TP and TM for colorectal cancer (CRC) primary and liver metastasis cells, respectively.

**Table 1 ijms-22-04819-t001:** Functional classification of geromiRs compiled from three review articles. The colored geromiRs are specific for each publication.

Hallmark	Harries 2014 [26]	Ugalde et al. 2014 [27]	Caravia and López-Otín 2015 [28]
Altered DNA damage response	*miR-21, * *miR-24, miR-34a, miR-34b, miR-34c,* *miR-106b, miR-125b, miR-192, miR-194,* *miR-210, * *miR-215, * *miR-421, * *miR-504*	*miR-18a, miR-22,* *miR-24, miR-29, miR-34a, miR-34b, miR-34c, * *miR-99, miR-138, miR-182, * *miR-210, miR-373, miR-421, * *miR-605*	*let-7, miR-1, miR-16-1, * *miR-29, * *miR-34, miR-103, miR-124, miR-143, miR-145*
Loss of telomeres	*miR-34a, miR-34b, miR-34c, miR-138, * *miR-155*		*miR-155, miR-200, miR-498*
DNA methylation	*miR-9, miR-29a, miR-29b, miR-29c, miR-34a, miR-34b, miR-34c, miR-124a, miR-127, miR-143, miR-148a, miR-152, miR-200*		
Histone modifications	*miR-15a, miR-16, miR-26a, miR-29a, miR-29b, miR-29c, miR-98, miR-101, miR-144*		
Regulation of splicing	*miR-1, miR-7, miR-10a, miR-10b, miR-16, miR-124a, miR-125a, miR-137, miR-193a-3p, miR-340, miR-519*		
Changes to protein homeostasis	*miR-1, miR-26b, miR-106b, miR-301b, miR-320*		*miR-9, miR-16, miR-17-5p, miR-34, miR-101, miR-130, miR-376b, miR-2016a, miR-E1108, miR-E1016*
Altered nutrient sensing	*miR-1, miR-17, miR-19b, miR-20a, miR-106a, * *miR-126, miR-190b, * *miR-206, miR-320,* *miR-486*		*miR-1, miR-17, miR-19b, miR-20a, miR-106a, * *miR-145, miR-182, * *miR-206, * *miR-223, * *miR-320, * *miR-470, miR-669b, miR-681*
Mitochondrial dysfunction	*miR-34a, * *miR-145, * *miR-146a, * *miR-335*		*let-7b, miR-19b, miR-20b,* *miR-34a, * *miR-34b, miR-34c, miR-106a, miR-133b, * *miR-146a, * *miR-181a, miR-221*
Cellular senescence	*let-7a, miR-17, miR-19b, * *miR-20a, * *miR-29a, miR-34a, miR-34b, miR-34c, miR-106a, miR-217, miR-369-3p, miR-371, miR-372, miR-373, miR-499*	*miR-20a, miR-24, miR-146*	*let-7, miR-21, miR-26b, miR-33, miR-181a, miR-210, miR-424*
Stem cell exhaustion	*let-7a, miR-29c, miR-290, miR-291-3p, miR-292-3p, miR-293, miR-294, miR-295, miR-371, miR-369-3p, miR-499*		*let-7b, miR-25, miR-33, miR-93, miR-106b, miR-141-3p, miR-486-5p, miR-489, miR-598*
Inflammaging	*miR-21, miR-146a, miR-155*		
Epigenetic alterations (Sirtuins)		*miR-9, miR-34, miR-135a, * *miR-181, * *miR-199b, miR-204, miR-217, * *miR-486, * *miR-519*	*miR-9, miR-34, miR-135a, * *miR-191a, miR-191b, * *miR-199b, miR-204, miR-217, * *miR-290,* *miR-519*
Stem cell homeostasis		*let-7b, miR-33, miR-106b-25, miR-302, miR-486, miR-489, miR-598*	
Insulin/IGF1		*lin-4, miR-1, miR-71, miR-145, miR-206, miR-239, miR-320, miR-470, miR-669b, miR-681*	
Altered Intercellular Communication			* let-7, miR-21, miR-29a, miR-71, miR-80 *

**Table 2 ijms-22-04819-t002:** Functional classification of geromiRs compiled from three review articles that are represented in the used microarrays. The significantly downregulated geromiRs are in bold red.

Hallmark	GeromiRs
Altered DNA damage response	***miR-106b*** *, miR-125b, miR-138, miR-182, * ***miR-18a*** *, miR-192,* ***miR-194*** *, miR-210, miR-215, * ***miR-22***, ***miR-24*** *, miR-34a, miR-34b, miR-34c, miR-421, miR-504*
Loss of telomeres	***miR-103****, miR-124, miR-138,****miR-143***, ***miR-155****, miR-34a, miR-34b, miR-34c*
DNA methylation	*miR-127, ****miR-143****, miR-148a,****miR-152***, ***miR-29a***, ***miR-29b***, ***miR-29c****, miR-34a, miR-34b, miR-34c, miR-9*
Histone modifications	*miR-144, ****miR-15a***, ***miR-16***, ***miR-26a***, ***miR-29a***, ***miR-29b***, ***miR-29c****, miR-98*
Regulation of splicing	***miR-10a*** *, miR-10b, miR-125a, miR-137, * ***miR-16*** *, miR-340*
Changes to protein homeostasis	***miR-106b***, ***miR-26b****, miR-301b, miR-320*
Altered nutrient sensing	*miR-106a, ****miR-126****, miR-17, miR-182, miR-190b, ****miR-19b****, miR-206,****miR-20a***, ***miR-223****, miR-320, miR-470, miR-486, miR-669b, miR-681*
Mitochondrial dysfunction	*let-7b, miR-106a, miR-133b, ****miR-146a****, miR-181a,****miR-19b***, ***miR-20b****, miR-221, ****miR-335****, miR-34a, miR-34b, miR-34c*
Cellular senescence	*let-7a, miR-106a, miR-17, miR-181a, ****miR-19b***, ***miR-20a****, miR-210, miR-217,****miR-24***, ***miR-26b***, ***miR-29a****, miR-33, miR-34a, miR-34b, miR-34c, miR-499*
Stem cell exhaustion	*let-7a, let-7b,****miR-106b***, ***miR-25****, miR-290, miR-293, miR-294, miR-295, ****miR-29c****, miR-33, miR-489, miR-499, miR-598,****miR-93***
Inflammaging	***miR-146a*** *, **miR-155***
Epigenetic alterations (Sirtuins)	*miR-135a, miR-199b, miR-204, miR-217, miR-290, miR-486, miR-9*
Stem cell homeostasis	*let-7b,* ***miR-16*** *, miR-33, miR-376b, miR-486, miR-489, miR-598, miR-9*
Insulin/IGF1	*miR-206, miR-320, miR-470, miR-669b, miR-681*
Altered Intercellular Communication	***miR-29a***

## Data Availability

The data discussed in this publication have been deposited in NCBI’s Gene Expression Omnibus (GEO) [66] and are accessible through GEO Series accession number GSE156431 [66] (https://www.ncbi.nlm.nih.gov/geo/query/acc.cgi?acc=GSE156431 accessed on 2 November 2020).

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
