# Peer review of "GeromiRs Are Downregulated in the Tumor Microenvironment during Colon Cancer Colonization of the Liver in a Murine Metastasis Model"

_ijms, 2021, doi:10.3390/ijms22094819_

Round 1

Reviewer 1 Report

The manuscript entitled “ GeromiRs are down-regulated in tumor microenvironment during colon cancer colonization of liver in murine metastasis model” investigates the change of specific geromiRs in the liver tumor microenvironment (TME) produced by colon cancer. From the 115 geromiRs and their associated hallmarks of aging, which authors compile from the literature, 75 are represented in the used microarrays and 26 out of them are down-regulated in the TME cells during colon cancer colonization of liver and none of them are up-regulated. The histone modification hallmark of the down-regulated geromiRs is significantly enriched with the geromiRs miR-15a, miR-16, miR-26a, miR-29a, miR-29b, miR-29c. Authors built a network of all the down-regulated in the TME cells geromiRs and their gene targets from the MirTarBase database, and authors analyzed the expression of these geromiR gene targets in the TME. Authors found that Cercam and Spsb4, identified as prognostic markers in some cancer types, are upregulated in the TME cells.

    In summary, liver TME cells show downregulation of geromiRs that promote cancer progression and TME cells phenotypic transformation to cancer supporting like-cells. Additionally, the most important downregulated geromiRs are clustered in epigenetic modulations such as histone modification, DNA methylation and sirtuins modulation. MicroRNAs that are upregulated during aging process are downregulated in the TME showing again the similarities between aging and cancer but in the opposite way resembling the two sides of a coin.

English language and style check are required.

Author Response

Dear Mr. George Danalache, Assistant Editor of International Journal of Molecular Science

Thank you very much for your mail of April 17th about your decision on the manuscript ijms-1177600 entitled "GeromiRs are down-regulated in tumor microenvironment during colon cancer colonization of liver in murine metastasis model".

Please find enclosed the revised version of the manuscript. For the preparation of this new version, we have carefully taken into account the new comments from the Referees. We have greatly appreciated these comments. The changes made in the manuscript in response to these reviewers' suggestions have been highlighted using red color font. In addition, we include our comments to the reviewers´ concerns and to the aforementioned changes in a point-by-point fashion. During correction, the authors detected parts of the manuscript that required better explanation or new bibliographic references to strengthen the main idea, and they have been added; these new text fragments and citations also are highlighted.  We now hope that the manuscript is suitable for publication in the International Journal of Molecular Science. If you have further questions, please do not hesitate to contact me.

With kind regards,

Iker Badiola

------------------------------------------------------------------------------------

ANSWERS TO THE REVIEWERS´ COMMENTS

Journal: International Journal of Molecular Science

Manuscript ID: ijms-1177600

Title: "GeromiRs are down-regulated in tumor microenvironment during colon

cancer colonization of liver in murine metastasis model ".

Author(s): Daniela Gerovska, Patricia Garcia-Gallastegi, Olatz Crende, Joana Márquez, Gorka Larrinaga, Maite Unzurrunzaga, Marcos J. Araúzo-Bravo , Iker Badiola.

Reviewer #1:

Comments and Suggestions for Authors

The manuscript entitled “GeromiRs are down-regulated in tumor microenvironment during colon cancer colonization of liver in murine metastasis model” investigates the change of specific geromiRs in the liver tumor microenvironment (TME) produced by colon cancer. From the 115 geromiRs and their associated hallmarks of aging, which authors compile from the literature, 75 are represented in the used microarrays and 26 out of them are down-regulated in the TME cells during colon cancer colonization of liver and none of them are up-regulated. The histone modification hallmark of the down-regulated geromiRs is significantly enriched with the geromiRs miR-15a, miR-16, miR-26a, miR-29a, miR-29b, miR-29c. Authors built a network of all the down-regulated in the TME cells geromiRs and their gene targets from the MirTarBase database, and authors analyzed the expression of these geromiR gene targets in the TME. Authors found that Cercam and Spsb4, identified as prognostic markers in some cancer types, are upregulated in the TME cells.

    In summary, liver TME cells show downregulation of geromiRs that promote cancer progression and TME cells phenotypic transformation to cancer supporting like-cells. Additionally, the most important downregulated geromiRs are clustered in epigenetic modulations such as histone modification, DNA methylation and sirtuins modulation. MicroRNAs that are upregulated during aging process are downregulated in the TME showing again the similarities between aging and cancer but in the opposite way resembling the two sides of a coin.

English language and style check are required.

Answer

We appreciate the reviewer comment. The main requirement is focused on language and style. The manuscript has been reviewed form the language point of view. The corrections are highlighted in red font. Grammatical, orthographic, and typographic errors have been corrected.

Reviewer 2 Report

The manuscript by Gerovska and colleagues reports the results of an in vivo study aimed at investigating the differential gene expression in normal tissue and TME of mice xenotransplanted with a murine CRC cell line. The main findings show how several genorMRs (mostly involved in epigenetic phenomena) were differentially expressed in the presence of cancer cells.

The paper is interestingly, methods are appropriate and results are sound. However, I would know whether the presence of liver metastases was demonstrated in mice. There are not information about this point in the manuscript.

Author Response

Dear Mr. George Danalache, Assistant Editor of International Journal of Molecular Science

Thank you very much for your mail of April 17th about your decision on the manuscript ijms-1177600 entitled "GeromiRs are down-regulated in tumor microenvironment during colon cancer colonization of liver in murine metastasis model".

Please find enclosed the revised version of the manuscript. For the preparation of this new version, we have carefully taken into account the new comments from the Referees. We have greatly appreciated these comments. The changes made in the manuscript in response to these reviewers' suggestions have been highlighted using red color font. In addition, we include our comments to the reviewers´ concerns and to the aforementioned changes in a point-by-point fashion. During correction, the authors detected parts of the manuscript that required better explanation or new bibliographic references to strengthen the main idea, and they have been added; these new text fragments and citations also are highlighted.  We now hope that the manuscript is suitable for publication in the International Journal of Molecular Science. If you have further questions, please do not hesitate to contact me.

With kind regards,

Iker Badiola

------------------------------------------------------------------------------------

ANSWERS TO THE REVIEWERS´ COMMENTS

Journal: International Journal of Molecular Science

Manuscript ID: ijms-1177600

Title: "GeromiRs are down-regulated in tumor microenvironment during colon

cancer colonization of liver in murine metastasis model ".

Author(s): Daniela Gerovska, Patricia Garcia-Gallastegi, Olatz Crende, Joana Márquez, Gorka Larrinaga, Maite Unzurrunzaga, Marcos J. Araúzo-Bravo , Iker Badiola.

Reviewer #2:

Comments and Suggestions for Authors

The manuscript by Gerovska and colleagues reports the results of an in vivo study aimed at investigating the differential gene expression in normal tissue and TME of mice xenotransplanted with a murine CRC cell line. The main findings show how several genorMRs (mostly involved in epigenetic phenomena) were differentially expressed in the presence of cancer cells.

The paper is interestingly, methods are appropriate and results are sound. However, I would know whether the presence of liver metastases was demonstrated in mice. There are not information about this point in the manuscript.

Answer

We also appreciate reviewer 2 comments. Following the suggestion of reviewer 1, the reviewer 2 also asked for a language revision, which has been done as we explain in the previous answer.

Concerning the second point related with the demonstration of liver metastasis, pictures of mice livers have been added to the Figure 1, in order to demonstrate the correct performance of liver metastasis experiment.
